# Flavor Preference and Systemic Immunoglobulin Responses in E-Cigarette Users and Waterpipe and Tobacco Smokers: A Pilot Study

**DOI:** 10.3390/ijerph17020640

**Published:** 2020-01-19

**Authors:** Monica Jackson, Kameshwar P. Singh, Thomas Lamb, Scott McIntosh, Irfan Rahman

**Affiliations:** Department of Environmental Medicine and Public Health Sciences, University of Rochester Medical Center, School of Medicine & Dentistry, Rochester, NY 14642, USA; mjacks29@u.rochester.edu (M.J.); Kameshwar_Singh@URMC.Rochester.edu (K.P.S.); Thomas_LambJr@URMC.Rochester.edu (T.L.); Scott_McIntosh@URMC.Rochester.edu (S.M.)

**Keywords:** e-cigarette, flavors, respiratory symptoms, immune response, biomarkers, motivation

## Abstract

Electronic cigarette (e-cigarette) use has had an exponential increase in popularity since the product was released to the public. Currently, there is a lack of human studies that assess different biomarker levels. This pilot study attempts to link e-cigarette and other tobacco product usage with clinical respiratory symptoms and immunoglobulin response. Subjects completed surveys in order to collect self-reported data on tobacco product flavor preferences. Along with this, plasma samples were collected to test for immunoglobulin G (IgG) and E (IgE) levels. Our pilot study’s cohort had a 47.9% flavor preference towards fruit flavors and a 63.1% preference to more sweet flavors. E-cigarette and traditional cigarette smokers were the two subject groups to report the most clinical symptoms. E-cigarette users also had a significant increase in plasma IgE levels compared to non-tobacco users 1, and dual users had a significant increase in plasma IgG compared to non-tobacco users 2, cigarette smokers, and waterpipe smokers. Our pilot study showed that users have a preference toward fruit and more sweet flavors and that e-cigarette and dual use resulted in an augmented systemic immune response.

## 1. Introduction

Currently, toxicity from e-cigarette exposure has been observed in acellular, cellular, and human models. E-cigarette vapors have been shown to generate acellular reactive oxygen species (ROS), and exposure to e-cigarette vapors results in an increased generation of ROS by small airway epithelial cells, which poses a potential for lung injury [1]. E-liquid exposure to mucosal tissue has resulted in cytotoxicity and an increase in DNA fragmentation, and exposure to e-cigarette vapors have resulted in cytotoxicity and DNA strand breaks in epithelial cells [2,3]. E-cigarettes use is perceived to be safer than traditional cigarette smoking as there are fewer carcinogens, but concerns regarding pulmonary and cardiovascular diseases remain [4]. Short-term e-cigarette use was found to have immediate adverse effects on pulmonary function, with an increase in lung impedance and peripheral airway flow resistance and a decrease in fractional exhaled nitric oxide [5]. Cardiovascular risks of e-cigarette use are found to be lower than the risks associated with traditional cigarette smoking, but they may pose a great risk to individuals with a predisposition to cardiovascular disease [6]. 

Given the diversity of e-cigarette products, there has been research exploring the mechanisms in which e-cigarettes affect the body, specifically the immune system. Innate defense proteins, such as elastase and matrix metalloproteinase-9, secreted in the airways are altered in the sputum of e-cigarette users when compared to nonsmokers and have resulted in both similar and unique alterations compared to traditional cigarette smoking [7]. In a rat model, exposure to tobacco smoke resulted in an increase in serum immunoglobulin E (IgE) levels, but did not alter immunoglobulin G (IgG) and immunoglobulin M serum levels [8]. Similar increases in IgE levels were seen in a general population of adult cigarette smokers [9]. It was also observed that ex-smokers had a decrease in levels of IgE once smoking ceased, and the spike of IgE seen in humans due to seasonal allergic rhinitis in non-smokers, but was not seen in smokersthough smokers had high levels of serum IgE [9]. However, to the best of our knowledge, there have been no studies with a diverse group of human subjects that associate e-cigarette use with immunoglobulin level. Currently, only research with mouse models are available; for example, BALB/c mice sensitized to ovalbumin and exposed to e-cigarette vapors experienced increased IgE and cytokine levels compared to unexposed mice [10]. Recently, a survey-based study showed that e-cigarette users reported allergic rhinitis responses; however, immune responses by flavors were not studied [11].

It is widely accepted that chronic obstructive pulmonary disease (COPD) can be linked with cigarette smoking and that the most effective management for COPD is smoking cessation [12,13]. Chronic smoking is associated with a higher percentage of symptoms like cough, shortness of breath, and other respiratory problems [14]. Given the similar toxic effects from cigarette smoke and e-cigarette vaping in cellular models, it should then be tested whether or not similar results in immune-inflammatory responses occur. Because e-cigarettes have only been on the market for a short time, long-term effects have not yet been studied. However, clinical symptoms from e-cigarette use, like coughing and dry or irritated mouth or throat, are prominent [15].

This pilot study aimed to determine the flavor preference of tobacco product users within our cohorts. We also attempted to determine the clinical symptoms that tobacco product users have, as well as to determine whether tobacco product usage can potentially induce an immune-inflammatory response by measuring immunoglobulin levels, such as IgE [16]. 

## 2. Materials and Methods

### 2.1. Ethics Statements: Institutional Biosafety Approvals

All experiments performed in this pilot study were approved and in accordance with the University of Rochester Institutional Biosafety Committee. These protocols were approved by the Institutional Review Board (IRB) at the University of Rochester Medical Center, Rochester, N.Y. Written informed consent was obtained from all study participants.

### 2.2. Scientific Rigor Statement

We used a rigorous and unbiased approach throughout the experimental plans and when analyzing the data to ensure that our data would be reproducible, with full and detailed reporting of both methods and analyzed data. All of the key chemical resources used in this pilot study were validated, authenticated, and of a scientific standard from commercial sources. Our results adhere to the National Institutes of Health (NIH) standards of reproducibility and scientific rigor.

### 2.3. Subjects/Participants

This was a cross-sectional pilot study with data collected from 2016–2019, using self-reporting measures on a variety of information in regards to product usage and symptoms. Subjects were recruited via local newspaper and magazine advertisements in the Greater Rochester Area. 

The pilot study was conducted at the University of Rochester Medical Center (URMC), New York (IRB approval #RSRB00064337 and IRB approval #RSRB00063526), via the Clinical Research Center (CRC). The participants were selected based on a self-reported questionnaire containing information about demographic variables, clinical symptoms, electronic cigarette use, and vaping history and behavior. The participants were categorized into six groups between two separate cohorts. Cohort 1: Non-tobacco users 1, individuals who do not use any tobacco products (*n* = 26) and e-cigarette users (*n* = 22); and cohort 2: Non-tobacco users 2 (*n* = 25), cigarette smokers (*n* = 26), waterpipe smokers (*n* = 12), and dual smokers, comprising both waterpipe and cigarette smokers (*n* = 10). Details on inclusion and exclusion criteria have been reported previously [17,18]. Briefly, inclusion for cohort 1 was based on age, and exclusion was based on other tobacco product usage, individuals with chronic illnesses, or individuals currently infected with pulmonary/respiratory pathogens. Exclusion for cohort 2 was based on age and usage and individuals with chronic illnesses or currently infected with pulmonary/respiratory pathogens. Female participants currently breast feeding or pregnant were also excluded [17,18]. Subjects’ self-reported use was biochemically confirmed via plasma cotinine assay. 

### 2.4. Measures of Self-Reporting

All subjects were asked to fill out a survey based on the type of tobacco product used. The surveys had all the same questions, and wording only varied on the type of product used in the questions. Data generated for each group were based on responses about the subject’s usage, motivation, and quitting methods. 

### 2.5. Collection and Processing Blood Samples

Collection was done following previous methods. Briefly, whole venous blood was collected from participants, and plasma was processed and separated by centrifugation for 10–15 min at 2000× *g* within 60 min of collection [17,18].

### 2.6. Immunoglobulin (IgE and IgG) Quantification by ELISA

Plasma levels of IgE were quantified in samples collected from e-cigarette users, non-tobacco users 1, non-tobacco users 2, smokers, waterpipe smokers, and dual users using an ELISA kit (Invitrogen, Carlsbad, CA, USA, BMS2097) following manufacturer instructions with the minor modification of using a five-fold dilution for non-tobacco users 1 plasma samples and a ten-fold dilution for e-cigarette-user, non-tobacco users 2, smoker, waterpipe, and dual-user plasma samples. 

Plasma levels of IgG were quantified in samples collected from e-cigarette users, non-tobacco users 1, non-tobacco users 2, smokers, waterpipe smokers, and dual users using an ELISA kit (Sigma, St. Louis, MO, USA, RAB0001) following manufacturer instructions.

### 2.7. Statistical Analysis

We used the outcome (IgE and IgG) levels for subjects enrolled in *n* = 25 with baseline (i.e., non-tobacco users) and analyzed for power calculations using the calculated mean and standard deviations from the data. Using the proc power procedure in SAS v9.4 (SAS Institute Inc., Cary, NC, USA), we computed the power of our statistical analysis through an exact method with a significance level alpha of 0.05. We found that the power ranged from 0.05 to 0.152 for our IgE data analysis and from 0.05 to 0.313 for our IgG data analysis. Analyses used to explore statistically significant differences included using either an unpaired *t*-test or one-way ANOVA with Tukey’s post hoc test for multiple comparisons by GraphPad Prism Software version 8.1.1. The results are shown as mean ± SD. Outliers were tested using RUOT (robust regression and outlier removal) with Q = 1% by GraphPad Prism Software version 8.1.1. Data were considered to be statistically significant for *p* values < 0.05.

## 3. Results

### 3.1. Description of Subjects

Cohort I: In our cohort 1, consisting of non-tobacco users 1 and e-cigarette users, the average age of non-tobacco users 1 was 33.88 ± 14.07 years old and consisted of 42.30% males and 57.69% females. The average age of e-cigarette users was 35.54 ± 12.21 years old and consisted of 45.45% males and 54.54% females. In non-tobacco users 1, the demographic breakdown was 69.23% Caucasian, 11.53% African American, 15.39% Asian, and 3.84% Hispanic. The e-cigarette users’ demographic breakdown was 50.00% Caucasian, 27.27% African American, 13.63% Asian, and 9.09% Hispanic. Non-tobacco users 1 had no history of e-cigarette vaping or smoking, while e-cigarette users had a vaping duration of 2.00 ± 1.64 years and no history of smoking [18]. 

Cohort II: In our cohort 2, consisting of non-tobacco users 2, cigarette smokers, waterpipe smokers, and dual users, the average age of non-tobacco users 2 was 36.16 ± 12.52 years old and consisted of 52.00% males and 48.00% females. The average age of cigarette smokers was 46.73 ± 9.96 years old and consisted of 50.00% males and 50.00% females. The average age of waterpipe smokers was 33.16 ± 14.61 years old and consisted of 66.66% males and 33.33% females, and the average age of dual users was 39.50 ± 12.49 years old and consisted of 60.00% males and 40.00% females. In non-tobacco users 2, the demographic breakdown was 84.00% Caucasian, 4.00% African American, and 12.00% Asian. The cigarette smokers’ demographic breakdown was 61.54% Caucasian, 30.76% African American, 3.84% Asian, and 3.84% Hispanic. The waterpipe smokers’ demographic breakdown was 41.66% Caucasian, 25.00% African American, and 33.33% Asian. The dual users’ demographic breakdown was 70.00% Caucasian, 10.00% African American, and 20.00% Asian. Non-tobacco users 2 had no history of cigarette smoking or waterpipe smoking. Cigarette smokers had a duration of 20.03 ± 8.55 years of cigarette smoking and no history of waterpipe smoking. Waterpipe smokers had a duration of 2.72 ± 1.84 years of waterpipe smoking and no history of cigarette smoking. Finally, dual smokers (both cigarette and waterpipe) had a duration of 14.00 ± 12.89 years of cigarette smoking and a duration of 4.69 ± 3.72 years of waterpipe smoking. 

### 3.2. Vaping Patterns in E-Cigarette Users

Smoking patterns in e-cigarette users were measured by self-reporting in three ways: Duration, frequency, and session length. Of the three categories, the most common responses were a duration less than half a year, frequency of greater than ten sessions per day, and a session length of less than five minutes. However, responses varied from half a year to greater than five years’ duration, a frequency of fewer than three sessions per day to greater than ten, and a session length of less than five minutes to a maximum of twenty-minute sessions (data not shown).

### 3.3. Flavor Preferences of All Subjects

Nearly half (47.9%) of all subjects preferred fruit flavors, with tobacco being the next most popular flavor (25%), followed by menthol (16.67%), spices (6.25%), and candy (4.16%) (Figure 1A). From this list, it was determined that 63.16% of subjects preferred sweeter flavors over flavors that were less sweet (Figure 1B). For e-cigarette users specifically, 70% of males preferred more sweet flavors, while 63.63% of female users preferred less sweet flavors, more sweet flavors being defined as those that have a sweeter taste, such as those in the fruit and candy categories. For both genders, at least 50% reported that their most preferred flavor was fruit (data not shown).

### 3.4. Clinical Symptoms of Tobacco Product Users

The most common clinical symptom reported by e-cigarette users had eleven subjects reporting a cough 1–2 times per week and only two e-cigarette users reporting having any chest pain/tightness. The most common clinical symptom reported by traditional cigarette smokers had nine subjects reporting a cough several times per week. Cigarette smokers also had the most subjects reporting chest pain/tightness, with four subjects reporting chest pain/tightness 1–2 times per week. Most waterpipe smokers reported having no clinical symptoms with only two subjects reporting chest pain/tightness and another two subjects reporting a cough. Similar to both smokers and waterpipe smokers, thee dual-user subjects reported either a cough several times per week, a cough 1–2 times per week, or chest pain/tightness 1–2 times per week (Table 1).

### 3.5. Plasma Immune Biomarkers: IgE and IgG Levels

E-cigarette users had a significant increase in plasma IgE levels when compared to non-tobacco users 1. (Figure 2A). However, there was no significant change in plasma levels of IgG in e-cigarette users compared to non-tobacco users 1 (Figure 2B). There was also no significant difference between non-tobacco users 2, smokers, waterpipe smokers, or dual users in plasma IgE levels (Figure 2C). However, in dual users, there was a significant increase in plasma IgG levels compared to non-tobacco users 2, smokers, and waterpipe smokers (Figure 2D).

## 4. Discussion

This pilot study attempted to relate e-cigarette and other tobacco product use and plasma immunoglobulin levels with users’ self-reported clinical symptoms in order to bring more attention to the need for longitudinal research. Our pilot study showed a significant elevation in IgE levels (a product of allergic reaction) in e-cigarette users compared to non-tobacco users 1, and e-cigarette users reported the second-highest clinical symptoms of all groups, behind only traditional cigarette smokers. A majority of e-cigarette users reported coughing one to two times per week compared to coughing several times a week among traditional cigarette smokers. This, however, could be due to the shorter duration of vaping e-cigarettes compared to traditional cigarette smokers. Similar to our results, another study conducted on wheezing risk of adult e-cigarette users found that vaping had an increased risk in both wheezing and other related respiratory symptoms [11]. 

Although there was no significant increase in plasma IgE levels between smokers and non-tobacco users 2, previous papers have found significant increases in IgE levels in adult smoking populations [9]. Smoker plasma IgG levels were found to be decreased in our cohort, although not significant, which is in line with other research that has shown that an increase in the number of cigarettes smoked would result in a decrease in serum IgG levels [19]. Even though there was a significant increase in plasma levels of IgG in dual-users, there were three values that were higher than the majority, while not being outliers. These values may have driven the average to become significantly higher than the other subject groups. Due to this being a pilot project, our current sample size is small, and with a larger subject group from current subject recruitment, we may be able to determine if the select values are driving the average higher than it should be. 

E-cigarettes have been marketed to adults as a smoking cessation aid or as a safe alternative nicotine delivery system. However, data are mixed on whether or not they are truly helpful [20]. A prospective cohort study in 2015–2016 that looked at whether or not electronic nicotine delivery systems (ENDS) aided in smoking cessation in adult smokers found that ENDS use did not result in higher quitting rates in adult smokers compared to smokers who did not use ENDS [21]. 

Our e-cigarette subjects reported experiencing clinical symptoms due to their use, although there were no significant effects in lung function of e-cigarette users. The reason there was no alteration in lung function is unknown, but previous studies have shown that the immediate effects of e-cigarette use result in adverse effects on pulmonary function [5]. Regardless of the presence of clinical symptoms, a majority of e-cigarette users reported believing that e-cigarettes are very safe to use. A study looking at college students’ perception of safety for e-cigarette use found that individuals that used e-cigarettes perceived e-cigarette use as safe and posing no risk of second- or third-hand effects [22]. Even though e-cigarette users perceived e-cigarettes as safe, it was shown that e-cigarette users had an elevation in other pro-inflammatory mediators [18]. Perception of e-cigarette health effects are inconsistent with the fact that, even in the short term, users are experiencing adverse effects. A greater focus should be placed on not only conducting research on the health consequences of e-cigarette use, but also on making these findings readily available for the general public for risk assessment and regulation.

Our cohort of subjects showed a near majority preference for fruit-flavored products and a majority preference for sweeter flavors. This is in line with another study that looked at JUUL flavor usage in middle-schoolers and high-schoolers, which showed that fruit-flavored pods like mango and fruit were preferred by this young cohort [23]. Despite the fact that e-cigarettes were first introduced to the market in 2007, there has been a rapid market expansion with the amount of unique flavors more than doubling from 2013–2014 to 2016–2017, with 15,568 distinct flavors being sold [24]. With this continuously changing product, future studies should be conducted in a longitudinal model with a larger cohort, looking at immunoglobulin, pro-inflammatory, and oxidative stress biomarkers in e-cigarette users and non-users, with a specific interest in the flavoring effects on these biomarkers. Furthermore, due to recent episodes of e-cig (or vaping) product use-associated injuries, it may be worthwhile to include these cohorts for determining systemic immune-inflammatory responses associated with clinical symptoms.

Despite the ability of this pilot study to correlate biological samples with clinical symptoms reported by a group of subjects that use a diverse variety of tobacco products, there are certain limitations to the pilot study. One example is the small sample size due to the limited amount of time collecting samples in the Monroe County. Further work is in progress to increase the sample size in various demographics, using the nationwide Population Assessment of Tobacco and Health (PATH) study samples to correlate the immune-inflammatory response with tobacco flavorings. 

## 5. Conclusions

Our pilot study shows a potential link between clinical symptoms and usage of e-cigarettes and other tobacco products. Our pilot study also showed elevated plasma IgE levels in e-cigarette users compared to non-tobacco users 1, potentially indicating a significant immune response in e-cigarette users. Clinical symptoms for e-cigarette users were the second-most prevalent for subjects, except for subjects that used traditional cigarette smoking. Future studies are required to study subjects over a period of time in a longitudinal study to assess long-term immune response based on flavors and their clinical symptoms.

## Figures and Tables

**Figure 1 ijerph-17-00640-f001:**
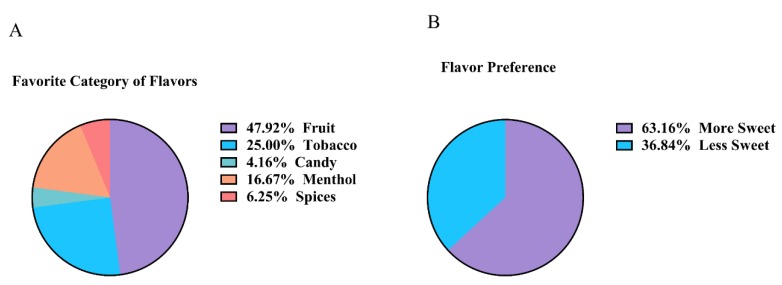
Flavor preferences of subjects. (**A**) Self-reported survey data based on favorite flavor. Total number of subjects = 48. (**B**) Self-reported survey data based on sweetness of flavor. Total number of subjects = 38.

**Figure 2 ijerph-17-00640-f002:**
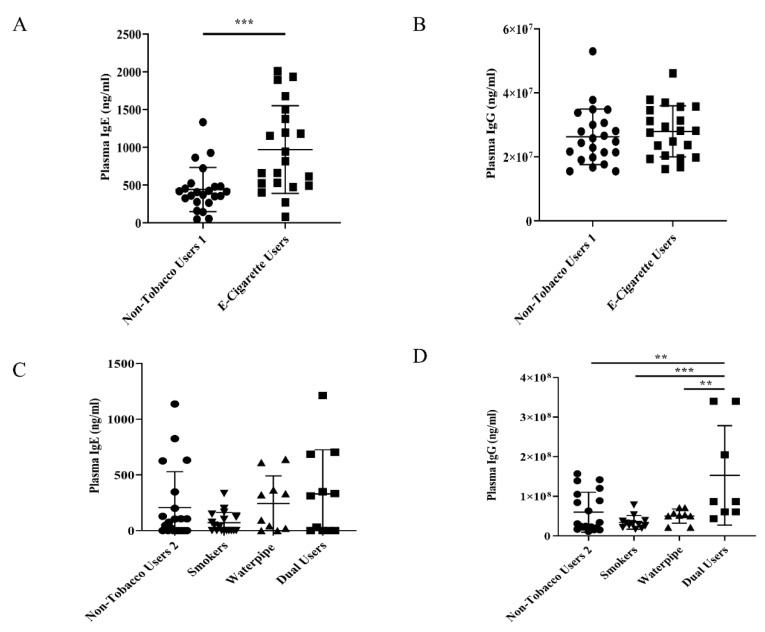
Plasma immunoglobulin levels in tobacco product users. (**A**) Plasma samples collected from either e-cigarette users or non-tobacco users 1 were measured for IgE levels following a 1:5 dilution or 1:10 dilution, respectively. *** *p* < 0.001 vs. non-tobacco users 1. N = 21–23. Outliers removed following robust regression and outlier removal test (RUOT). (**B**) Plasma samples collected from either e-cigarette users or non-tobacco users 1 were measured for IgG levels following a 1:10,000,000 dilution. N = 21–23. Outliers removed following RUOT test. (**C**) Plasma samples collected from either non-tobacco users 2, smokers, waterpipe smokers, or dual users were measured for IgE levels following 1:10 dilution. N = 11–21. Outliers removed following RUOT test. (**D**) Plasma samples collected from either non-tobacco users 2, smokers, waterpipe smokers, or dual users were measured for IgG levels following a 1:10,000,000 dilution. ** *p* < 0.01, *** *p* < 0.001 vs. indicated subjects. N = 8–20. Outliers removed following RUOT test, and certain samples were out of range of the plate reader.

**Table 1 ijerph-17-00640-t001:** Clinical symptoms experienced by e-cigarette and tobacco users.

	Cough Every Day	Cough Several Times/Week	Cough < 1 Time/Week	Cough 1–2 Times/Week	Chest Pain/Tightness Several Times/Week	Chest Pain/Tightness 1–2 Times/Week	None
E-Cigarette	1	3	1	11	1	1	5
Waterpipe	1	1	0	0	1	1	9
Traditional Cigarettes	3	9	0	0	0	4	0
Dual Users (Traditional cigarettes and waterpipe)	0	3	0	3	0	3	1

Classification of chest pain and cough are based on frequency of symptom experience. Values are reported based on survey data as a total number experiencing that criteria. Varying sample sizes between variables is due to omission of some self-reported items by some subjects.

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
