# Peer review of "Flavor Preference and Systemic Immunoglobulin Responses in E-Cigarette Users and Waterpipe and Tobacco Smokers: A Pilot Study"

_ijerph, 2020, doi:10.3390/ijerph17020640_

Round 1
Reviewer 1 Report
Peer Review Manuscript ID: ijerph-655028, Flavors and systemic immunoglobulin responses in E-cigarette users, and waterpipe and tobacco smokers, by Monica Jackson et al.
General comments:
The study aims add collecting subject self-reported data on use patterns, clinical symptoms, flavor preferences, quitting aids and safety perceptions of different smokeable tobacco/non-tobacco products, including dual-users, along with the quantification of plasma IgG and IgE. The authors conclude on e-cigarette flavor preferences, clinical symptoms, and plasma IgG and IgE levels.
It is indeed very important to assess the health effects linked to e-cigarette use, and therefore solid scientific studies should be produced. Their rigor is a pre-requisite for providing regulators the necessary information for decision making.
Despite the scientific rigor statement provided by the authors, this cross-sectional study suffers of major shortcomings. 1. Sample size: the sample investigated is small, while no motivation / statistical justification is provided for the choice of the sample. While the authors mention that a variety of product use patterns were reported, it is mostly probable that a great variability in the type of products used, as well as the intensity is confounding the outcome of the study. 2. Self-reporting of findings is always biased and requires larger cohorts. 3. Inclusion/exclusion criteria relative to medication used, physical and health condition are missing. Consulting the provided references (17 and 18), does not add additional information, especially since 18 is in press. In the case the e-cigarette users were former smokers, the time period of e-cigarette use is important to interpret the clinical findings, in other words, the wash out period is quite critical. 4. The immune response markers studies are very unspecific and the reviewer wonders about their added value. As the authors correctly stress, specific markers such as pre-inflammatory mediators, oxidative stress markers, and cardiac and lung function assessment should be performed in order to be conclusive.
Specific comments:
Introduction:
The authors seem to be quite biased in their selection of citations, selecting those that are reporting deleterious effects of e-vapor experiments. Briefly reviewing the non-clinical work presented reveals that very high concentrations of liquids/extracts have been used in the experimental settings. Their relevance is not commented. It is a basic principle in toxicology that everything is toxic – but it is a matter of dose… A more balanced approach would be appropriate, while weighing the scientific merit of all published work in the field. Line 2: ‘generate acellular reactive oxygen species’. While this terminology is used in the referenced paper, it doesn’t say much. What is really meant here? Investigations in cell-free systems? Study/Data transparency: is the study registered at clinicaltrials.gov ?Materials and Methods, participants.
Need a clear definition of non-users: what population is exactly meant here? What is the difference with non-smokers? Dual users: what is meant? E-cig and cigarette smokers? What is the proportion? What is the evidence that e-cigarette users do not use different products as well? Sample size, rationale (see general comments)Results
Plasma IgE and IgG levels: Fig 3: The reviewer is puzzled by the different y-axes used in A and C, while the same parameter has been measured. Between non-users and non-smokers, at least a factor 10 difference is found in the plasma IgE concentration is found, while between non-users and e-cigarette users approx. a factor 2 difference is found, which is reported s a positive effect. D. It is unclear how dual use (although not really defined) can lead to such large changes in IgG levels compared to the other conditions assessed. If as stated in the discussion, smoking is reducing IgG levels, what is the (biological) significance of the higher IgG levels?Conclusion
First sentence: the statement is incorrect: no relationship has been established, and the is based on one very variable and non-specific biomarker (see also remark 6). Not biomarkers, as stated. Mentioning an elevated allergic response is also no appropriate, this can only be stated if specific (relative to the allergen) IgEs are changing. Moreover, due to the variability of the results (non-smokers, vs non-users), it is a stretch to draw such conclusion. It is difficult to reconcile the fact that e-cigarette users have the highest prevalence of symptoms among those that do not use traditional cigarettes. If dual users use both, traditional cigarettes and e-cigarettes, wouldn’t it be fair to expect that the dual users are more affected than the e-cigarette users?Author Response
Attached

Reviewer 2 Report
Thank you for giving me the opportunity to review the manuscript entitled “Flavors and systemic immunoglobulin responses in E-cigarette users, and waterpipe and tobacco smokers”. The manuscript would be of interest for the journal however in current form it presents mixture of information and it is really difficulty to recognize what is the scope and rationale for the study
Please find below several comments which in my opinion can improve the manuscript:
The title needs to be modified – it need to correspond/reflect the aim of the study – in this version it is difficult to recognize about the scope of the paper – two different thing (in my opinion not really related to each other) were studied – maybe one (flavors) can be omitted The abstract of the study presents mixture of information – the authors need to decide what they want to present and modify the paper accordingly Some statistics need to be added into the abstract of the manuscript I would modify “human subject studies” into “human studies” The last paragraph of introduction contains repetitions – please modify it – to present clearly the aim of the study The sample size for each group is very small – the authors need to present the power calculation The number of approval (IRB approval # RSRB00064337 and IRB approval # RSRB00063526) can be moved from subjects/participants subsection into Ethics statements: Institutional biosafety approvals I prefer to present the subject characteristics (and it is the standard in scientific papers) in the form of table not figure – you can include the table in supplementary materials “Motivation for e-cigarettes”, “Quit attempts” and “Perception of tobacco product safety” in my opinion are not related to the aim of the study so they can be deleted Strengths and limitations of the study need to be added.Author Response
Attached

Reviewer 3 Report
Jackson and colleagues explored the impact of e-cigarette usage on respiratory symptoms by self-assessment and quantification of plasma immunoglobulin, total IgG and IgE. E-cigarette and traditional cigarette smokers reported the most clinical symptoms. Fruit flavored e-cigarettes were the most popular and e-cigarette users believed e-cigarettes were safe despite higher IgE levels demonstrated here. Individuals using 2 products had elevated IgG. This article addresses an important topic. However, there are several issues that require addressing. These are itemized below, not in order of significance, but in order of presentation.
Only age, gender, and ethnicity are reported in figure 1. We need to know more about the study population. How long are the subjects using the products, i.e. pack years? Equally, were any pulmonary diseases diagnosed in the subjects, e.g. asthmatics, COPD? Equally, demographics should be presented for each user group, i.e. e-cigarette user, tobacco user, etc. We need to see mean ages and S.D. for each group Although tables 1-3 are informative that could be presented in a better manner to make it easier for the reader to decipher the outcomes. Please perform data analysis on this data and present the data in a different manner. Maybe have variables in columns and categories in rows, rather than repeating categories 4 times Please define what “more sweet” means. What are the flavors in each group? Be specific Minor, please define RUOT test In figure 3, what is the definition of non-users? Are they nonsmokers or current tobacco smoke users? Equally, there appears to be a large difference in IgE and IgE levels between figures 3A vs 3C, and 3B vs 3D (at least a tenfold difference). Is this due to plate variation in assays? What is the range of detection of the ELISA?Author Response
Attached

Round 2
Reviewer 1 Report
Review of adapted manuscrit after first round of review.
The overall comments provided by the current reviewer remain true, because the scientific quality of the paper remains quite poor. In my opinion, the paper should be rejected, and the data combined with the other paper the authors mention relative to specific markers of inflammation. I do have the impression that the authors try to slice up all their data in different papers as to obtain more papers on the same subject. This fragmented approach is not providing a full overview on the study outcome and therefore not appropriate. Up to the editor to decide what she/he wants to decide on accepting the paper or not.
In case it is going to be accepted the following changes need to be implemented.
Despite the fact that the authors have added the words 'Pilot Study' in the title, this wording and the limitations addressed in the previous round of review should be added in the abstract, introduction and discussion of the paper. The replies given to the comments 1, 2, 3 are not satisfactory - the paper should be adapted taking the reviewer's concerns into consideration. relative to comment 4, previous revision: it is still unclear whether non-users and non-smokers are a similar type of population. I understand they are coming from 2 different cohorts, but using different names for probably similar profiles is confusing. the added piece in the section Statistical analyses is vague and should be supported by numbers. What was the statistical power that was calculated? The added piece of text on page 4 contains k lines 7 and 11. Explanation on dual user effect for IgG should be given. It seems that the mean for the dual users of cigarettes and water pipe are determined by 3 outliers. This should be commented and the relevance placed into perspective. Moreover, as questioned previously, how do the authors explain the obviously lower value for IgG in smokers, while smoking is know to be pro-inflammatory? In the conclusion, the following sentence is hard to undersatnd based on the data provided:'Clinical symptoms for e-cigarette users were the most prevalent for subjects that did not use traditional cigarettes.' There is no dual user group of e-cigs and traditional cigarettes in the study. Moreover the wash out period is not addressed - authors state that the e-cigarette users have never used cigarettes - is this really the case?Author Response
Manuscript ID: ijerph-655028 (Revised)
Title: Flavor preference and systemic immunoglobulin responses in E-cigarette users, and waterpipe and tobacco smokers: A pilot study
Authors: Monica Jackson, Kameshwar P. Singh, Thomas Lamb, Scott McIntosh and Irfan Rahman
Overall Response:
We would like to thank the editor and reviewer for allowing us to revise our manuscript, ‘Flavor preference and systemic immunoglobulin responses in E-cigarette users, and waterpipe and tobacco smokers: A pilot study’. We are grateful for the comments made by the reviewer and have attempted to fix the flaws pointed out by the reviewer in the small time allotted in order to create a better manuscript. We have highlighted the changes we have compared to the first revised manuscript. We would like to make note that we have altered Figure 2 in order to adhere to the new definition for non-users/non-smokers.
Review of adapted manuscript after first round of review.
The overall comments provided by the current reviewer remain true, because the scientific quality of the paper remains quite poor. In my opinion, the paper should be rejected, and the data combined with the other paper the authors mention relative to specific markers of inflammation. I do have the impression that the authors try to slice up all their data in different papers as to obtain more papers on the same subject. This fragmented approach is not providing a full overview on the study outcome and therefore not appropriate. Up to the editor to decide what she/he wants to decide on accepting the paper or not.
In case it is going to be accepted the following changes need to be implemented.
Despite the fact that the authors have added the words 'Pilot Study' in the title, this wording and the limitations addressed in the previous round of review should be added in the abstract, introduction and discussion of the paper. Response: We agree with the fragmentary theme, but we did this new study after the first study completed realizing the immune data and perception of flavors in our cohorts. We are not trying to slice up the work. We have now revised the manuscript based on this reviewer’s above comments. We have addressed this issue and added in this wording into the abstract, introduction, and the discussion in the paper
The replies given to the comments 1, 2, 3 are not satisfactory - the paper should be adapted taking the reviewer's concerns into consideration.
Response: We had attempted to complete as much revisions and reply as best as we could during the short time allotted for our first round of revisions. In which we removed sections of the manuscript, added in descriptive background on the subjects collected, reformatted tables, added a more descriptive inclusion/exclusion criteria, reworded sections of the manuscript all based on the suggestions of the last round of comments.
Relative to comment 4, previous revision: it is still unclear whether non-users and non-smokers are a similar type of population. I understand they are coming from 2 different cohorts, but using different names for probably similar profiles is confusing.
Response: Each group are of similar populations since both consist of subjects that have never used any tobacco products, and since there seems to be confusion about the subjects we have decided to use a similar name of non-tobacco users but with the distinction of 1 or 2 to determine the cohort they were collected from.
The added piece in the section Statistical analyses is vague and should be supported by numbers. What was the statistical power that was calculated? The added piece of text on page 4 contains k lines 7 and 11.
Response: We have added in more specifics about our power calculation in the manuscript in order to clear up any vagueness to our original statement.
Explanation on dual user effect for IgG should be given. It seems that the mean for the dual users of cigarettes and water pipe are determined by 3 outliers. This should be commented, and the relevance placed into perspective.
Response: Despite the fact the three values may look like outliers, we have performed outlier tests on all values and these values are not outliers, but we have now commented about the three high values and addressed the potential relevance.
Moreover, as questioned previously, how do the authors explain the obviously lower value for IgG in smokers, while smoking is known to be pro-inflammatory?
Response: Although cigarette smoking is known to be pro-inflammatory, it is also known in the literature that cigarette smoking results in a decrease in IgG levels. We have cited a paper that showed this result previously in our discussion on page 5 lines 14 to line 16.
In the conclusion, the following sentence is hard to understand based on the data provided: 'Clinical symptoms for e-cigarette users were the most prevalent for subjects that did not use traditional cigarettes.' There is no dual user group of e-cigs and traditional cigarettes in the study.
Response: That sentence is in reference to Table 1, in which the e-cigarette subjects had the second most self-reported clinical symptoms, with only traditional cigarette smokers had the most reported clinical symptoms. So we have altered the wording so that the intention of the sentence is clearer.
Moreover the wash out period is not addressed - authors state that the e-cigarette users have never used cigarettes - is this really the case?
Response: E-cigarette users self-reported they have not used cigarettes, and biomarkers were measured to determine smoking history. The washout period would have been six months as per the questionnaires.